# Advances in the Diagnosis and Treatment of Pediatric Acute Lymphoblastic Leukemia

**DOI:** 10.3390/jcm10091926

**Published:** 2021-04-29

**Authors:** Hiroto Inaba, Ching-Hon Pui

**Affiliations:** 1Department of Oncology, St. Jude Children’s Research Hospital, Memphis, TN 38105, USA; ching-hon.pui@stjude.org; 2Department of Pediatrics, University of Tennessee Health Science Center, Memphis, TN 38163, USA

**Keywords:** acute lymphoblastic leukemia, pediatric, advances, diagnosis, treatment

## Abstract

The outcomes of pediatric acute lymphoblastic leukemia (ALL) have improved remarkably during the last five decades. Such improvements were made possible by the incorporation of new diagnostic technologies, the effective administration of conventional chemotherapeutic agents, and the provision of better supportive care. With the 5-year survival rates now exceeding 90% in high-income countries, the goal for the next decade is to improve survival further toward 100% and to minimize treatment-related adverse effects. Based on genome-wide analyses, especially RNA-sequencing analyses, ALL can be classified into more than 20 B-lineage subtypes and more than 10 T-lineage subtypes with prognostic and therapeutic implications. Response to treatment is another critical prognostic factor, and detailed analysis of minimal residual disease can detect levels as low as one ALL cell among 1 million total cells. Such detailed analysis can facilitate the rational use of molecular targeted therapy and immunotherapy, which have emerged as new treatment strategies that can replace or reduce the use of conventional chemotherapy.

## 1. Introduction

Approximately 6000 new cases of acute lymphoblastic leukemia (ALL) are diagnosed in the United States annually [1,2,3,4]. ALL is the most common pediatric cancer (representing approximately 25% of cancer diagnoses), and approximately 60% of all cases occur in children and adolescents younger than 20 years, with an annual incidence of 36.2 per 1 million persons and a peak age of incidence of two to five years (at which there are >90 cases per 1 million persons) [5]. ALL is diagnosed more frequently in boys than in girls, with a ratio of approximately 1.3:1. The annual incidence of ALL differs markedly according to race and ethnic group; there are 40.9 cases per million in the Hispanic population, 35.6 cases per million in the white population, and 14.8 cases per million in the black population [6]. ALL cases are broadly classified as B-ALL or T-ALL based on immunophenotyping, with B-ALL comprising approximately 85% of cases, although this percentage can differ depending on age at diagnosis, race, or ethnicity.

Currently, the survival of pediatric patients with ALL treated in high-income countries exceeds 90% (Figure 1) [1,2,3,4]. Chemotherapy is given in four important phases: remission induction, consolidation, reinduction (delayed intensification), and continuation (maintenance). Chemotherapy is administered based on stratified risk classification, as determined by clinical factors (e.g., age (1–9.9 years vs. <1 or ≥10 years) and white blood cell (WBC) counts (<50 × 10^9^/L vs. ≥50 × 10^9^/L) at diagnosis), cytogenetic and genomic analysis of ALL cells, and response evaluation with a minimal residual disease (MRD) assay. Dosage adjustment based on pharmacodynamic and pharmacogenomic studies and supportive care (e.g., prevention and treatment of infection) have also contributed substantially to improved outcomes. Therefore, current dosages/schedules for “conventional” chemotherapy have been truly optimized.

With the current high rate of survival, further improvement in outcomes with conventional chemotherapy is challenging. In fact, there was very little improvement in 5-year overall survival (OS) between our two recent frontline ALL trials, St. Jude Total Therapy XV (5-year OS: 93.5%) and XVI (5-year OS: 94.3%) (Figure 1) [7,8]. Most of the conventional chemotherapy agents were approved by the US Food and Drug Administration before 1980 (Table 1), and their therapeutic intensity has been pushed to the limit of tolerance. Accordingly, further intensification of conventional chemotherapy could lead to only minimal improvement in overall outcomes while increasing adverse effects.

Recently, several molecular targeted agents and immunotherapy approaches have been introduced, and they promise to improve outcomes. For these agents to be used optimally, detailed genetic characterization of leukemia cells and response evaluation by MRD in individual patients are critical. In this review, we will review the genetic subgroups of ALL, the evaluation of MRD, and newer treatment strategies.

## 2. Genetic Characterization of Acute Lymphoblastic Leukemia

The revolutionized approach to genomic analysis subdivides pediatric ALL into more than 30 genetic subgroups [9,10,11]. In B-ALL, recurrent genomic subtypes are characterized by chromosomal aneuploidy, i.e., hyperdiploidy (>50 chromosomes) or hypodiploidy (<44 chromosomes), and by rearrangements: *ETV6/RUNX1* fusion, *TCF3/PBX1* fusion, *BCR/ABL1* fusion, and *KMT2A* (*MLL*) rearrangement (Figure 2 and Table 2). Genetic abnormalities newly identified by comprehensive genomic analyses include *BCR/ABL1*-like ALL (Ph-like ALL), intrachromosomal amplification of chromosome 21 (iAMP21), *DUX4*-rearranged ALL, *ZNF384*-rearranged ALL, *MEF2D*-rearranged ALL, *PAX5*-altered (PAX5alt) ALL, *NUTM1*-rearranged ALL, and *ETV6/RUNX1*-like ALL. Characterization of genetic abnormalities in ALL cells is important in order to identify unfavorable genetic abnormalities and to incorporate molecular targeted therapy to reduce the risk of relapse.

## 3. Low-Risk Genetic Subgroups

### 3.1. ETV6/RUNX1-Rearranged ALL

*ETV6/RUNX1*-rearranged ALL represents approximately 20% of pediatric ALL and is associated with excellent outcomes [13]. Up to 5% of normal newborns carry the *ETV6/RUNX1* fusion at birth [14], and post-natal environmental or spontaneous oncogenic second hits are required to induce overt leukemia [15,16]. Patients with the *ETV6/RUNX1* fusion are good candidates for reductions in the intensity of chemotherapy if their initial MRD responses are good [17,18]. A randomized study of patients with standard-risk ALL enrolled on the Associazione Italiana di Ematologia e Oncologia Pediatrica–Berlin-Frankfurt-Münster (AIEOP–BFM) ALL 2000 protocol tested whether dose reductions by 30% for dexamethasone and by 50% for vincristine, doxorubicin, and cyclophosphamide during the delayed intensification phase resulted in outcomes comparable to those in the historical arm [19]. Although this study led to worse outcomes for the dose-reduction arm as a whole, outcomes in patients with *ETV6/RUNX1* fusion and in those aged 1 to 6 years were equivalent for the two arms. Furthermore, in the Tokyo Children’s Cancer Study Group L92-13 study, which featured only 1 year of intensive chemotherapy, only two-thirds of the enrolled patients experienced continuous remission, but those with *ETV6/RUNX1* and *TCF3/PBX1* rearrangements had excellent outcomes with this abbreviated therapy [20]. Notably, patients with high hyperdiploidy fared poorly in this study.

### 3.2. Hyperdiploid ALL

Hyperdiploid ALL is the most common subtype of ALL, accounting for up to 25% of pediatric ALL. Different study groups have variously identified this subtype as having a DNA index of 1.16 or higher [21], a chromosome number of 51 to 67 [22], or trisomy of chromosomes 4 and 10 (double trisomy) [23]. Non-random gains of chromosome 4, 10, 14, 17, and 21 are common. Methotrexate is particularly useful for treating this subtype of ALL, and the disease response is influenced by the intracellular accumulation of active methotrexate polyglutamate metabolites (MTXPGs), which is higher in hyperdiploid ALL than in *ETV6/RUNX1* ALL, *TCF3/PBX1* ALL, or T-ALL [24,25,26]. This is partly due to the higher expression of the gene encoding the folate influx transporter *SLC19A1* in hyperdiploid ALL, resulting from the presence of a somatically acquired additional chromosome 21 on which this gene is located. Therefore, among patients with induction failure, those with hyperdiploid ALL had better outcomes than did those in other subgroups because they responded well to high-dose methotrexate, which is typically given as post-induction therapy, and these patients can be salvaged even without a hematopoietic cell transplant (HCT) [27].

Patients with *ETV6/RUNX1* fusion and hyperdiploidy and negative MRD on day 15 (as in St. Jude Total Therapy XVI) or day 19 (as in Total Therapy XV) and at the end of induction therapy have an excellent prognosis [11,17,18]. In St. Jude Total Therapy studies, patients with *ETV6/RUNX1* fusion and hyperdiploidy are provisionally treated in the low-risk (National Cancer Institute [NCI] standard-risk) arm regardless of their age or WBC count at diagnosis, but those patients with high MRD levels on day 15 (≥1%) or at the end of induction therapy (≥0.01%) or with extramedullary (central nervous system or testis) involvement are subsequently treated in the standard-risk (NCI high-risk) arm. This approach has been successful, with excellent outcomes for both subgroups [11,13,17].

### 3.3. DUX4-Rearranged ALL

*DUX4*-rearranged ALL is a newly identified subtype seen in 3% to 5% of pediatric ALL cases. The rearrangement occurs most commonly in the immunoglobulin heavy-chain locus (*IGH*) and results in the expression of DUX4 protein with a truncated C-terminus [28,29,30]. This truncated form binds an intragenic region of the ETS-family transcription factor *ERG* (ETS-related gene) and commonly results in the expression of a C-terminal ERG protein fragment that is a dominant-negative inhibitor of wild-type ERG function. *DUX4*-rearranged B-ALL has a unique immunophenotype (CD2 and CD371 positive), and a favorable outcome can be obtained, even with the deletion of *IKZF1*, by adjusting the intensity of the chemotherapy based on the MRD [31,32].

## 4. High-Risk Genetic Subgroups in B-ALL

### 4.1. Hypodiploid ALL

Hypodiploid ALL, which is defined by there being fewer than 44 chromosomes or a DNA index of less than 0.81, accounts for 1% to 2% of pediatric ALL. It is associated with poor outcomes, with reported EFS of 50% to 55% [33,34]. It can be classified into three distinct subtypes: near haploid (24 to 31 chromosomes), low hypodiploid (32 to 39 chromosomes), and high hypodiploid (40 to 43 chromosomes). Near-haploid ALL is associated with Ras pathway mutations (particularly in *NF1*) and *IKZF3* deletion [35]. Low-hypodiploid ALL is characterized by *TP53* mutations in the leukemia cells in more than 90% of cases and also in the germline in approximately 50% of patients, in addition to the somatic alterations in *IKZF2* and *RB1*. Therefore, patients with low-hypodiploid ALL should undergo germline testing for *TP53* germline pathogenic variants (i.e., Li–Fraumeni syndrome) to enable treatment modification to avoid the use of carcinogenic agents and for genetic consultation purposes [36]. It is important to distinguish “masked” hypodiploid ALL, in which the hypodiploid clone is duplicated, from true hyperdiploid ALL, considering the possible germline *TP53* mutations and the poor prognosis of hypodiploid ALL [37]. Recently, two multicenter studies demonstrated that HCT confers no benefit in hypodiploid ALL, particularly for patients who are MRD negative after remission-induction therapy, for whom EFS was approximately 70% [33,34]. Therefore, patients with persistently positive MRD can be considered for treatment with molecular targeted agents such as BCL-2 inhibitors and PI3K inhibitors or with immunotherapy such as bispecific antibody therapy or chimeric antigen receptor (CAR) T-cell therapy [35,38,39].

### 4.2. BCR/ABL1 (Philadelphia Chromosome)-Positive ALL

*BCR/ABL1*-positive ALL accounts for approximately 2% to 3% of pediatric ALL [40]. Before tyrosine kinase inhibitors (TKIs) became available, the survival of patients who were treated only with conventional chemotherapy was dismal, and HCT from a matched related donor or an unrelated donor during the first remission provided a cure in only approximately 50% of children [41]. The combination of the TKI imatinib with multi-agent chemotherapy significantly improved outcomes, with 5-year disease-free survival increasing to 70% in the Children’s Oncology Group AALL0031 study [42]. A second-generation TKI, dasatinib, targets both the ABL1 and SRC kinases, has activity against BCR/ABL1 that is approximately 300 times more potent than that of imatinib, and can cross the blood–brain barrier [40]. The Children’s Oncology Group AALL0622 study, in which dasatinib was used at 60 mg/m^2^/day, showed no improvement in outcomes relative to those in the preceding AALL0031 study, in which imatinib (340 mg/m^2^/day) was given with the same chemotherapy backbone [43]. However, the Chinese Children’s Cancer Group has shown that patients who received dasatinib (80 mg/m^2^/day) had significantly better EFS and OS and a lower relapse rate when compared with those who received imatinib (300 mg/m^2^/day) in a randomized study [44]. Ponatinib is one of a newer generation of TKIs and has potent activity in both wild-type *BCR/ABL1*-positive ALL and mutant forms (e.g., with the gatekeeper mutation ABL1 T315I) [40]. Treatment with ponatinib in combination with hyperfractionated cyclophosphamide, vincristine, doxorubicin, and dexamethasone (hyper-CVAD), alternating with high-dose methotrexate and cytarabine, resulted in excellent 2-year EFS in adults with newly diagnosed *BCR/ABL1*-positive ALL [45]. Using ponatinib in combination with a pediatric regimen that includes asparaginase and glucocorticoids can be difficult as ponatinib is also associated with an increased risk of thrombosis and pancreatitis. In adult patients with *BCR/ABL1*-positive ALL, a chemotherapy-free regimen with glucocorticoid and dasatinib followed by blinatumomab and dasatinib was associated with a high molecular response and high survival rates with few adverse effects [46]. Nevertheless, the results of a recent preclinical study suggest that dasatinib may adversely affect the efficacy of blinatumomab [47]; additional studies are needed to determine whether these two agents should be used separately.

### 4.3. BCR/ABL1 (Philadelphia Chromosome)-Like ALL

*BCR/ABL1* (Philadelphia chromosome)-like ALL was initially identified as a subgroup of leukemias with a leukemic cell gene expression profile similar to that of *BCR/ABL1*-positive ALL and frequent *IKZF1* alterations but without the *BCR/ABL1* fusion [48,49]. Although the percentage can vary depending on the ethnicity of the patients, this variant occurs in approximately 3% of pediatric ALL cases and is associated with worse outcomes. It is commonly seen in patients with NCI high-risk disease; however, *BCR/ABL1*-like ALL is also seen in patients with NCI standard-risk disease, and the outcome is associated with the MRD levels during and at the end of induction [50,51]. Many study groups have identified the genetic lesions associated with *BCR/ABL1*-like ALL, and these are classified in three main groups: JAK-STAT signaling activating mutations, *ABL1*-class fusions, and alterations that are less common and that involve other kinases [52,53].

JAK-STAT signaling activating mutations constitute the largest group and are genetically more diverse [52,53]. *CRLF2* rearrangements (*P2RY8/CRLF2* and *IGH/CRLF2*) and mutations (CRLF2 F232C) lead to CRLF2 overexpression, which may be detected by flow cytometry, and these mutations are present in approximately half of *BCR/ABL1*-like ALL cases, being more commonly seen in patients with Native American ancestry. Most of the *JAK1* and *JAK2* mutations are seen in this group. Other JAK-STAT signaling activating mutations are present in approximately 10% of *BCR/ABL1*-like ALL cases and include *JAK2* fusions (translocations or interstitial deletions) that retain the tyrosine kinase domain, *EPOR* truncating rearrangements (e.g., with *IGH*, *IGK,* and *LAIR1*), *IL7R* insertion/deletions in the transmembrane domain, and deletions or mutations of *SH2B3* (a negative regulator of JAK-STAT signaling, the mutation of which augments JAK-STAT signaling). A JAK inhibitor, ruxolitinib, is currently being tested in clinical trials [53].

*ABL1*-class fusions involve *ABL1*, *ABL2*, *CSF1R*, *PDGFRB*, and, rarely, *PDGFRA* and *LYN* and are seen in 15% to 20% of *BCR/ABL1*-like ALL cases [52,53]. Pediatric patients with *ABL1*-class fusions have poor outcomes when treated with regimens that do not contain a TKI, even when they receive a high-risk chemotherapy regimen and/or HCT [54]. As seen in *JAK2* fusions, these are chimeric in-frame fusions that preserve the tyrosine kinase domain and are, therefore, sensitive to treatment with ABL1 inhibitors such as imatinib and dasatinib [52,55].

Other rare kinase-activating alterations include those in *NTRK3*, *FLT3*, *PTK2B*, and *TYK2*, and preclinical studies have shown the efficacy of treatment of these variants with a TRK inhibitor, an FLT3 inhibitor, an FAK inhibitor, and a TYK2 inhibitor, respectively [52,53].

### 4.4. KMT2A-Rearranged ALL

The *KMT2A* (*MLL*) gene is located on chromosome 11q23 and can be rearranged with more than 80 different partner genes, which are seen in both lymphoid and myeloid leukemia [56]. *KMT2A*-rearranged ALL is characterized by the CD10-negative pro-B cell phenotype with co-expression of myeloid markers. It accounts for approximately 5% of pediatric ALL and 75% of infant ALL. In infant ALL, *KMT2A* rearrangement is acquired in utero and is associated with dismal outcomes, especially in infants younger than 6 months at diagnosis with a presenting WBC count of ≥300 × 10^9^/L or with a poor prednisone response [56]. Although two international randomized studies were performed to examine standard vs. more intensive therapy before maintenance therapy (the Interfant-99 study) and myeloid- vs. lymphoid-type consolidation therapy (the Interfant-06 study), there were no significant differences in outcomes between interventions or studies [57,58]. *KMT2A* rearrangement results in the assembly of a unique multi-protein complex with DOT1L, BRD4, and menin [59]. Therefore, there is great potential for molecular targeted therapy with inhibitors of DOT1L, bromodomain, menin, and BCL-2. Immunotherapy with blinatumomab and autologous or allogeneic CAR T cells can be considered, although there is a possibility of a lineage switch to acute myeloid leukemia (AML) [56].

### 4.5. MEF2D-Rearranged ALL

*MEF2D*-rearranged ALL is seen in approximately 1% of pediatric ALL cases. The *MEF2D* gene can rearrange with several partner genes: *BCL9* (the most common partner), *CSF1R*, *DAZAP1*, *HNRNPUL1*, *SS18*, and *FOXJ2* [9,10]. *MEF2D*-rearranged ALL is characterized by older age at diagnosis (median, 14 years), mature B-cell leukemia morphology (large, densely basophilic, and heavily vacuolated leukemic blasts), a unique immunophenotype (weak or absent expression of CD10, high expression of CD38, and cytoplasmic immunoglobulin μ-chain), and poor outcome due to early relapse [60,61,62]. Exogenous expression of *MEF2D/BCL9* in a B-ALL cell line promoted cell growth, increased the expression of HDAC9 (a known *MEF2D* target), and induced resistance to dexamethasone [60]. Patient-derived leukemia cells were sensitive to histone deacetylase inhibitors (vorinostat and panobinostat) and to a proteasome inhibitor (bortezomib) in vitro and in xenograft models. *MEF2D/CSF1R* can be targeted by ABL1 inhibitors.

### 4.6. TCF3/HLF-Rearranged ALL

*TCF3/HLF*-rearranged ALL is a rare (representing <0.5% of cases) but very aggressive subtype of ALL. It is mostly resistant to conventional chemotherapy and has extremely poor outcomes even with intensified chemotherapy and HCT [63]. *TCF3/HLF*-rearranged ALL is characterized by enrichment of stem cell and myeloid gene signatures, *PAX5* and *VPREB1* deletions, and Ras pathway gene mutations. *TCF3/HLF*-rearrangement plays a role as a pioneer transcription factor in recruiting EP300 to drive MYC, and EP300 inhibition reduces *TCF3/HLF*-dependent gene expression and ALL growth [64]. Drug activity profiling and preclinical studies have shown striking activity of a BCL-2 inhibitor, venetoclax [63]. Furthermore, all of nine patients with *TCF3/HLF*-rearranged ALL experienced molecular remission after being treated with blinatumomab, and four of them are in long-term remission after HCT, suggesting that an immunotherapy approach can overcome the chemotherapy resistance [65].

## 5. Intermediate-Risk Genetic Subtypes in B-ALL

### 5.1. TCF3/PBX1-Rearranged ALL

*TCF3/PBX1*-rearranged ALL is generated with the t(1;19)(q23;p13) translocation and is present in approximately 2% to 5% of pediatric ALL cases, commonly expressing cytoplasmic μ chain (a pre-B phenotype) [66]. As with *ETV6/RUNX1*-rearranged ALL, the preleukemic *TCF/PBX1* gene fusion is present in approximately 0.6% of healthy newborns [67]. The incidence of this leukemia variant is higher in African Americans [68], and a genome-wide association study identified a germline risk locus in an intergenic region between *BCL11A* and *PAPOLG*: rs2665658 [69]. In the St. Jude Total XV study, which eliminated cranial irradiation, *TCF/PBX1*-rearranged ALL was associated with a higher incidence of CNS relapse but a lower incidence of hematologic relapse compared to other forms of B-ALL [7,66]. In patients treated in the Total XVI study, the incidence of CNS relapse was reduced as a result of the increased frequency of early intrathecal treatments [8]. In the TCCSG L92-13 study, *TCF3/PBX1*-rearranged ALL had excellent outcomes with 1 year of intensive chemotherapy from diagnosis [20].

### 5.2. Intrachromosomal Amplification of Chromosome 21 (iAMP21)

Intrachromosomal amplification of chromosome 21 (iAMP21) ALL is characterized by the presence of additional copies of a region of chromosome 21 that includes *RUNX1* (five or more copies per cell), and it can be associated with the germline Robertsonian translocation rob(15;21) [70,71]. iAMP21 ALL is seen in approximately 1% to 2% of pediatric ALL cases and is associated with older age (median, 9 years) and low WBC counts. Secondary cytogenetic and genetic changes include the gain of chromosome X, the loss or deletion of chromosome 7, *ETV6* and *RB1* deletions, and *SH2B2* inactivation through copy number-neutral loss of heterozygosity of chromosome 12q [72,73]. Patients with iAMP21 had dismal outcomes when treated with a low-intensity NCI standard-risk regimen [74,75]. Although intensified treatment has significantly improved the outcomes for these patients, their EFS remains inadequate at approximately 70%. Therefore, they can also be considered candidates for recently introduced novel therapies.

### 5.3. PAX5-Driven Subtypes: PAX5alt and PAX5 p.Pro80Arg

PAX5 is the B-lymphoid transcription factor that is essential for early stages of B-cell development [76,77]. Germline alterations of the *PAX5* gene predispose patients to ALL, and somatic alterations of *PAX5* are commonly seen in pediatric ALL (e.g., *PAX5* focal deletions are present in approximately 30% of *ETV6/RUNX1*-rearranged ALL) [77]. The two distinct disease-initiating alterations of *PAX5* that result in PAX5alt and PAX5 p.Pro80Arg ALL account for approximately 3% to 5% and less than 1% of childhood ALL, respectively [9,10]. PAX5alt B-ALL is characterized by diverse *PAX5* alterations, including rearrangements (most commonly with *ETV6* or *NOL4L*), sequence mutations, and intragenic amplification. PAX5 p.Pro80Arg is characterized by universal p.Pro80Arg mutation with deletion or mutation of the remaining allele and alterations in Ras and JAK-STAT pathway genes. Patients with PAX5alt or PAX5 p.Pro80Arg B-ALL have an intermediate prognosis [9,10].

### 5.4. ZNF384-Rearranged ALL

*ZNF384*-rearrangement is seen in approximately 1% to 2% of childhood ALL cases and in half of B/myeloid mixed-phenotype acute leukemia (MPAL) cases in children. This rearrangement has more than 10 partner genes, such as *EP300*, *TCF3*, *TAF15*, and *CREBBP* [62,78,79]. In B-ALL, the age of onset and the prognosis differ according to the fusion partner: with the *EP300/ZNF384* fusion, the median age of onset is 11 years and outcomes are excellent, whereas with the *TCF3/ZNF384* fusion, the median age of onset is 5 years and there are occasional late relapses [78,80]. The immunophenotype of *ZNF384*-rearranged B-ALL is characterized by negative or weak expression of CD10 and aberrant expression of CD13 and/or CD33 [78,80]. As with *ETV6/RUNX1*-rearranged and *TCF3/PBX1*-rearranged B-ALL, a study in monozygotic twins showed that *TCF3/ZNF384* fusion can occur in utero, suggesting that a fetal hematopoietic progenitor is the cell of origin in this ALL subgroup [81]. Importantly, the secondary genomic alterations and gene expression profiles for *ZNF384*-rearranged B-ALL and B/myeloid MPAL cases are essentially indistinguishable, which suggests that ALL-directed therapy should be initiated for patients with newly diagnosed B/myeloid MPAL [79]. Due to its inherent lineage plasticity, *ZNF384*-rearranged leukemia may develop a lineage switch at relapse (from ALL to AML or vice versa) under the selective pressure of conventional chemotherapy or immunotherapy.

## 6. Other Newly Identified B-ALL Subtypes

### 6.1. ETV6/RUNX1-Like ALL

*ETV6/RUNX1*-like ALL is seen in 1% to 3% of pediatric ALL cases and is particularly common in younger children [9,10,30]. It has a similar gene expression profile and immunophenotype to *ETV6/RUNX1*-rearranged ALL but lacks the *ETV6/RUNX1* fusion. Within this group, alterations in *ETV6, IKZF1*, and *TCF3* have been reported. As the number of patients identified to date is small and several relapses have been reported, it is important to evaluate the actual outcomes of patients in this group, which appear to be worse than those of patients with *ETV6/RUNX1*-rearranged ALL.

### 6.2. NUTM1-Rearranged ALL

*NUTM1*-rearranged ALL is seen in 5% to 7% of all infants with ALL and represents 21.7% of non-*KMT2A*-rearranged infant ALL, but it is very rare in children (accounting for less than 1% in that population) [9,10,82,83]. Partner genes include *ACIN1*, *CUX1*, *BRD9*, and *ZNF618*. In an international study, the 4-year OS in 45 infants and 36 children was 100%, which is indicative of a favorable genetic subtype, although further studies are required to confirm this finding and to determine whether a reduction in treatment intensity is possible [82].

## 7. T-Acute Lymphoblastic Leukemia

T-ALL represents approximately 12% to 15% of pediatric ALL and is characterized by having an incidence in boys that is two to three times that in girls; a higher proportion of patients with African ancestry, in whom the rate is twice that in patients of European ancestry; high initial WBC counts; and higher frequencies of mediastinal mass and CNS involvement [12,84]. The higher incidence in boys can be partly explained by inactivating mutations or deletions of the tumor suppressor gene *PHF6* on chromosome X, which are seen in 16% of pediatric T-ALL cases [85]. The genetic alterations in T-ALL are diverse, and no clear associations with outcomes have yet been identified. Hence, unlike B-ALL, T-ALL lacks a consensus genetic classification with prognostic implications. In most cases of T-ALL, there is aberrant expression of transcription factors and oncogenes, including *TAL1*, *TAL2*, *LYL1*, *LMO1*, *LMO2*, *TLX1* (*HOX11*), *TLX3* (*HOX11L2*), and *HOXA* [86]. *NOTCH1* activating mutations and alterations in *CDKN2A/CDKN2B* are seen in more than 70% of cases, and *MLLT10* and *KMT2A* rearrangements are each seen in 5% of cases. Approximately 25% of patients have JAK-STAT activating mutations, and *ABL1* fusions with *BCR* and *NUP214* are occasionally detected [86]. These patients are candidates for treatment with JAK inhibitors and ABL1 inhibitors, respectively.

In most studies, the survival of patients with T-ALL is 5% to 10% worse than that of patients with B-ALL [12]. With regard to conventional chemotherapy, the treatment component of the BFM IB phase that includes cyclophosphamide, cytarabine, and mercaptopurine is of greater importance for T-ALL than for B-ALL [87]. In one study, patients with T-ALL who received nelarabine had significantly fewer incidences of CNS relapse (isolated and combined) when compared to patients who did not receive nelarabine [88]. However, approximately 90% of the total patients and all of the nelarabine-treated patients received cranial irradiation in this randomized study; therefore, the efficacy of nelarabine should be confirmed in patients whose disease is managed with intrathecal therapy only. The results of the recent randomized study of bortezomib are described below [89].

### Early T-Cell Precursor ALL

Early T-cell precursor (ETP) ALL accounts for 10% to 15% of T-ALL, having a specific immunophenotype of early T-cell development (cytoplasmic CD3+, CD5weak, CD8−, CD1a−) with aberrant expression of myeloid and/or early progenitor cell markers [90]. The genetic features of this subtype are similar to those of hematopoietic stem cells; it is characterized by alterations in transcriptional regulators, epigenetic regulation, and JAK-STAT and Ras pathway genes [86,91]. Furthermore, ETP ALL shares genomic features with T/myeloid MPAL, with frequent biallelic *WT1* alterations and signaling pathway mutations (e.g., in the JAK-STAT and *FLT3* pathways) [79]. ETP-ALL is usually glucocorticoid resistant, has a higher incidence of induction failure, especially after the BFM IA phase [92,93], and is historically associated with worse outcomes [90,94]. However, ETP-ALL responds to a regimen that includes cyclophosphamide, cytarabine, and mercaptopurine (e.g., the BFM IB phase), and its outcomes are approaching those of non-ETP T-ALL [92,93,95]. The results of a preliminary study suggested that patients with ETP-ALL would benefit from treatment with venetoclax, a BCL-2 inhibitor [96].

## 8. Minimal Residual Disease

Although genetic subclassification is essential for risk stratification, MRD has equally important prognostic and therapeutic impact [97,98,99]. MRD has been quantified by multiparametric flow cytometry or by allele-specific oligonucleotide PCR analysis. The flow cytometric assay uses the leukemia-specific aberrant immunophenotype, has a typical sensitivity of 0.01%, and can be applied to almost all cases of ALL [98,99]. It is rapid, enables accurate quantification of ALL cells, and provides an overview of the hematopoietic cell population status. However, it can be difficult to achieve sensitivity better than 0.01%, and the assay may fail to detect an ALL population that has undergone a phenotypic change, especially after immunotherapy targeting CD19 and/or CD22. The PCR assay amplifies leukemia-specific fusion transcripts (available for approximately 40% of ALL cases) or immunoglobulin (Ig) or T-cell receptor (TCR) genes (available for approximately 90% of ALL cases) with a sensitivity of 0.001%, 10 times that of the flow cytometry assay [98,99]. In RT-PCR analysis of fusion transcripts, there is a possibility of RNA degradation or cross-contamination from other samples. For Ig and TCR DNA, tailor-made primers are needed for each patient. Furthermore, ALL can be oligoclonal and may escape detection by clonal evolution during treatment. Recently, next-generation sequencing (NGS) of Ig or TCR genes has been applied for MRD detection (NGS MRD) with sensitivity as low as 0.0001% (equivalent to detecting one ALL cell among 1 million total cells) [100,101]. The use of universal primers enables the detection of clonal evolution and can also detect the background repertoire of normal B and T cells. With this technology, negative NGS MRD at the end of induction has been associated with 100% OS among NCI standard-risk patients [102]. In pediatric patients with ALL who received HCT, negative pre-HCT MRD and post-HCT MRD were associated with significantly fewer relapses and better survival [103]. The NGS MRD assay might not be affected by phenotypic changes after immunotherapy, and negative NGS MRD after CAR T-cell therapy was also associated with better outcomes as compared with those in patients with positive NGS MRD among the patients with negative flow MRD [104]. These clinical benefits will result in expanded use of NGS MRD in contemporary protocols.

When considering risk stratification, clinicians should consider MRD levels in combination with genetic classification and clinical factors (e.g., age, WBC counts at diagnosis, and lineage) [17,18,97,105]. Patients with favorable genetic features clear MRD faster than do those with unfavorable genetics or T-ALL. Furthermore, as seen in *ETV6/RUNX1*-rearranged and hyperdiploid ALL, some patients with favorable genetics but slow MRD clearance can be cured by intensifying their post-remission chemotherapy [11,17,27]. Conversely, patients with high-risk genetics have inferior outcomes even when they have undetectable MRD at the end of induction therapy [11,17,18]. It is also important to evaluate whether more sensitive NGS MRD can identify patients with better outcomes among those patients with high-risk genetic features. Furthermore, patients with T-ALL who had negative MRD (<10^3^) on day 78 had a cumulative risk of relapse similar to that of patients who had negative MRD on day 33 [87]. In such patients, the MRD level on day 33 was not relevant, suggesting that the MRD response to the BFM IB phase (two courses of cyclophosphamide, cytarabine, and mercaptopurine) is critical in T-ALL.

## 9. Emerging Therapy: Molecular Targeted Therapy

### 9.1. Tyrosine Kinase Inhibitors

Tyrosine kinase inhibitors have been employed in combination with standard chemotherapy to improve its efficacy (Table 1). As described earlier, ABL1 inhibitors (e.g., imatinib, dasatinib, nilotinib, and ponatinib) are used to treat patients with *BCR/ABL*-positive ALL and *ABL1*-class fusions that occasionally occur in *BCR/ABL*-like ALL and T-ALL [40,53,55]. Ruxolitinib is being tested in clinical trials for patients with JAK-STAT activating mutations as seen in *BCR/ABL*-like ALL and T-ALL (including ETP-ALL) [53]. Currently, however, this targeted approach is limited to less than 10% of pediatric ALL cases. Further identification of ALL driving mutations and their targets will expand the use of TKIs. In this regard, ex vivo leukemia drug-sensitivity profiling identified that 44.4% of childhood T-ALL samples and 16.7% of adult T-ALL samples as being sensitive to dasatinib through the inhibition of preTCR-LCK signaling [106].

### 9.2. BCL-2 and BCL-X_L_ Inhibitors

Members of the B-cell lymphoma 2 (BCL-2) protein family play critical roles in the intrinsic mitochondrial apoptosis pathway through interactions between pro- and anti-apoptotic proteins (Table 1) [107]. Venetoclax is a selective inhibitor of BCL-2 and displaces the pro-apoptotic proteins BIM and BAX, which leads to mitochondrial outer membrane permeabilization, cytochrome c release, and the activation of intracellular caspases, resulting in apoptosis. Preclinical studies have shown that venetoclax is active for leukemias in the high-risk genetic group, such as *KMT2A*-rearranged ALL [108], hypodiploid ALL [38], *BCR/ABL*-positive ALL [109], *TCF3/HLF*-rearranged ALL [63], and T-ALL (including ETP-ALL) [110,111]. Low expression of *CELSR2* is associated with the overexpression of *BCL2* and glucocorticoid resistance in ALL cells [112]. Venetoclax mitigated glucocorticoid resistance and had synergistic effects with prednisolone and dexamethasone.

Phase I studies of venetoclax in combination with chemotherapy in pediatric and young adult patients with ALL have shown the regimen to be well tolerated with preliminary efficacies [113]. As the results of a preclinical study suggested that ALL cells were dependent on both BCL-2 and BCL-X_L_, navitoclax (a BCL-2 and BCL-X_L_ inhibitor) was tested in combination with venetoclax and chemotherapy for pediatric and adult patients with relapsed/refractory ALL or lymphoblastic lymphoma [114]. Among 47 heavily pre-treated patients, the complete remission rate was 60%, showing the regimen to have promising efficacy.

### 9.3. Proteasome Inhibitors

Proteasome inhibitors have shown efficacy in ALL and work synergistically with chemotherapy agents such as corticosteroids and doxorubicin (Table 1) [115]. In 22 children with relapsed ALL treated with bortezomib in combination with vincristine, dexamethasone, pegaspargase, and doxorubicin, the overall response rate was 73% [116]. In a randomized study of patients with newly diagnosed T-ALL or T-lymphoblastic lymphoma (T-LLy), adding bortezomib to the induction and delayed intensification phases was associated with better outcomes, as compared to those in patients who did not receive bortezomib, in patients with standard-risk and intermediate-risk T-ALL, as well as in those with T-LLy [89]. However, addition of bortezomib was associated with worse outcomes in patients with high-risk T-ALL. Newer proteasome inhibitors (carfilzomib and ixazomib) are under investigation.

### 9.4. Other Molecular Targeted Therapies

Dysregulation of the PI3K/AKT/mTOR pathway is frequently observed in ALL and is associated with resistance to chemotherapy [117,118]. mTOR inhibitors have been shown to inhibit ALL growth and reverse glucocorticoid resistance and to work synergistically with other chemotherapeutic agents, such as dexamethasone, vincristine, and doxorubicin (Table 1) [119,120,121]. A phase I study of everolimus with vincristine, prednisone, pegasparagase, and doxorubicin in children and adolescents with ALL in first marrow relapse occurring more than 18 months after first complete remission showed that the regimen was tolerable [122]. Nineteen (86%) of 22 enrolled patients had a second complete remission, and 13 (68%) of them had negative MRD.

Epigenetic modification, the biochemical alteration of chromatin, has been implicated in the pathogenesis of cancer [123]. Instead of changes in the nucleotide sequence, epigenetic modifications involve DNA methylation and histone modification, which affect the activity of genes and their cellular expression. These modifications can silence tumor suppressor genes or activate oncogenes. They are prevalent in ALL and are associated with chemotherapy resistance and relapse [124]. Epigenetic modifications may be reversible with targeted agents such as DNA methyltransferase inhibitors and histone deacetylase inhibitors (Table 1). In a phase 1 study of decitabine and vorinostat in combination with vincristine, dexamethasone, mitoxantrone, and pegaspargase, 22 children and adolescents with relapsed or refractory ALL were treated [125]. Although this regimen was associated with a high incidence of infectious complications, nine patients (39%) had a complete response, and potent pharmacodynamic modulations of biological pathways associated with antileukemic effects were observed.

## 10. Emerging Therapy: Immunotherapy

Three major categories of immunotherapy are currently in use for pediatric ALL (Figure 3 and Table 1): bispecific antibodies (e.g., blinatumomab), CAR T cells, and antibody–drug conjugates (e.g., inotuzumab) [126]. Immunotherapy has been used mostly for B-ALL because the surface markers CD19, CD20, and CD22 are expressed only on B cells and not on hematopoietic stem cells or other tissues. Such therapy can eradicate not only B-ALL but also normal B cells, thereby causing hypogammaglobulinemia, which can be managed by intravenous or subcutaneous immunoglobulin administration. For T-ALL, antibody therapy (e.g., with daratumumab against CD38) and CAR T cells (e.g., anti-CD1a, CD5, and CD7) are under investigation (Figure 3).

### 10.1. Bispecific Antibody Therapy

Blinatumomab has bispecific single-chain Fv fragments that link CD3+ T cells to CD19+ leukemia cells and cause a cytotoxic immune response (Figure 3 and Table 1) [127,128]. It is approved for use in pediatric and adult relapsed/refractory and MRD-positive B-ALL by the US Food and Drug Administration. The main adverse effects are cytokine release syndrome and neurotoxicity, which coincide with T cell activation. Two randomized studies in children, adolescents, and young adults with intermediate-risk or high-risk relapsed/refractory B-ALL showed blinatumomab to have benefits over intensive consolidation chemotherapy [129,130]. The loss of CD19 expression is a major mechanism of resistance to blinatumomab treatment and is also observed with CAR T cell therapy. Acquired genetic mutations in *CD19* exons 2–5 or alternative splicing at exon 2 produce a truncated protein with a nonfunctional or absent transmembrane domain and/or no antibody binding site [131,132]. Sustained CD19-antibody pressure can result in lineage switches as described in *KMT2A*- and *ZNF384*-rearranged B-ALL [133,134]. An alteration in CD81, which is a chaperone protein for the maturation and trafficking of the CD19 molecule from the Golgi apparatus to the cell surface, has been also reported [135].

### 10.2. Chimeric Antigen Receptor (CAR) T Cells

CAR T cells express single-chain Fv fragments against B-lineage markers (e.g., CD19, CD22, or both) with intracellular signaling domains such as 4-1BB or CD28 with CD3ζ [136]. A phase 2 international study of anti-CD19 CAR T cells (tisagenlecleucel) in pediatric and young adult patients with relapsed/refractory B-ALL showed a complete remission rate of 81% at 3 months and EFS and OS of 73% and 90%, respectively, at 6 months [137]. Currently, tisagenlecleucel is approved for patients up to 25 years of age with B-ALL that is refractory or in a second or later relapse. Several groups consider CAR T cells to be curative therapy, although others view them as a bridging therapy to HCT. As with blinatumomab, cytokine release syndrome and neurotoxicity are commonly seen with CAR T-cell therapy [138]. Preemptive administration of tocilizumab (an anti-IL-6 receptor antibody) decrease the incidence of severe cytokine release syndrome without compromising the efficacy of CAR T cells [139]. CAR T-cell recipients are also at high risk for infection, and they should be considered for bacterial and fungal prophylaxis until their neutropenia resolves, in addition to immunoglobulin supplement and *Pneumocystis jirovecii* pneumonia prophylaxis [140].

Mechanisms of resistance to CAR T-cell therapy include the loss of CAR T-cell persistence and B-cell aplasia and antigen loss on ALL cells [141,142]. In the former scenario, the type of co-stimulatory molecule (e.g., 4-1BB vs. CD28), rejection due to the murine component in tisagenlecleucel, and T-cell exhaustion are considered important factors. The use of two co-stimulatory molecules or new types of co-stimulatory molecule; humanized CAR T cells; in vivo stimulation with a CD19 vaccine, cytokines, or check point inhibitors; or early collection of T cells during treatment for high-risk patients may overcome this issue. With regard to target antigen loss, CAR T cells that can target other antigens (e.g., CD22 or the thymic stromal lymphopoietin receptor) or that can simultaneously target dual antigens (e.g., CD19/CD22) and the administration of two independent CAR T cells that target different antigens are being investigated [143,144,145,146,147].

For extramedullary relapse (e.g., in the CNS and testes), CAR T cells can migrate and show anti-leukemia effects; therefore, they can be considered not only for isolated bone marrow relapses but also for isolated or combined extramedullary relapses, thereby avoiding radiation therapy [148,149].

### 10.3. Antibody-Drug Conjugates

Inotuzumab ozogamicin is an anti-CD22 antibody that is linked to calicheamicin, a cytotoxic antitumor antibiotic that causes double-strand DNA breaks [150]. Inotuzumab is currently approved for use in adult patients with relapsed/refractory B-ALL. It is associated with sinusoidal obstruction syndrome, especially after HCT [150]. Fractionated weekly dosing of inotuzumab at the dose lower than a single dose given every 3–4 weeks and a longer interval between inotuzumab administration and HCT (i.e., 2 months or more) can reduce the incidence of this syndrome [151]. Additionally, it is recommended to use prophylactic pharmacologic agents (e.g., ursodiol), to limit the inotuzumab use to two cycles if HCT is planned, and to avoid HCT conditioning regimens that contain dual alkylating agents (e.g., thiotepa and melphalan) and concomitant hepatotoxic drugs (e.g., azoles) [152]. In a pediatric phase I study that used fractionated weekly dosing for relapsed/refractory B-ALL, complete remission was seen in 80% of the patients and 84% of those with available flow cytometry data had negative MRD [153].

## 11. Conclusions

The diagnosis of ALL, the treatment of patients, and the evaluation of the treatment response have undergone remarkable improvement. The detailed genetic characterization of ALL cells, functional genomics and proteomics, and drug sensitivity assays with ex vivo and patient-derived xenograft (PDX) models for molecular targeted agents and immunotherapy will lead to new therapeutic strategies. Furthermore, the evaluation of germline genetics can lead to an understanding of leukemogenesis, cancer predisposition, and the differences in drug response and metabolism (pharmacogenomics). Basic, translational, and clinical research on ALL will not end until all patients can be cured without acute complications or late sequelae.

## Figures and Tables

**Figure 1 jcm-10-01926-f001:**
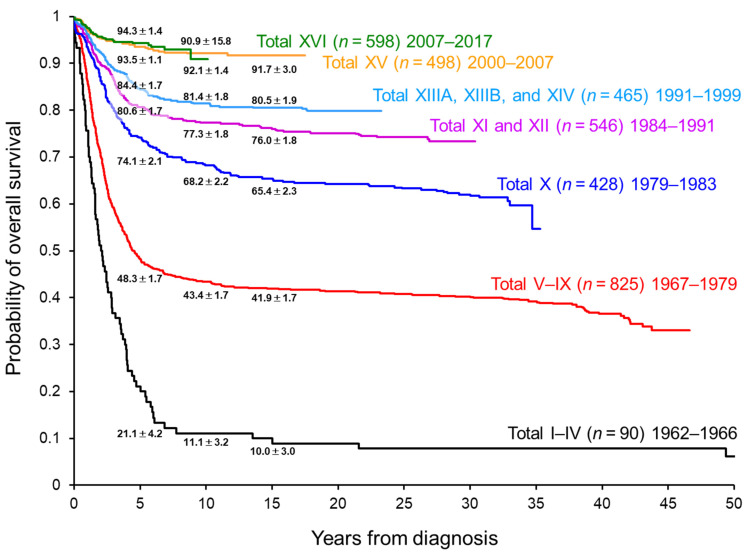
Overall survival of pediatric patients with acute lymphoblastic leukemia treated in the St. Jude Total Therapy studies.

**Figure 2 jcm-10-01926-f002:**
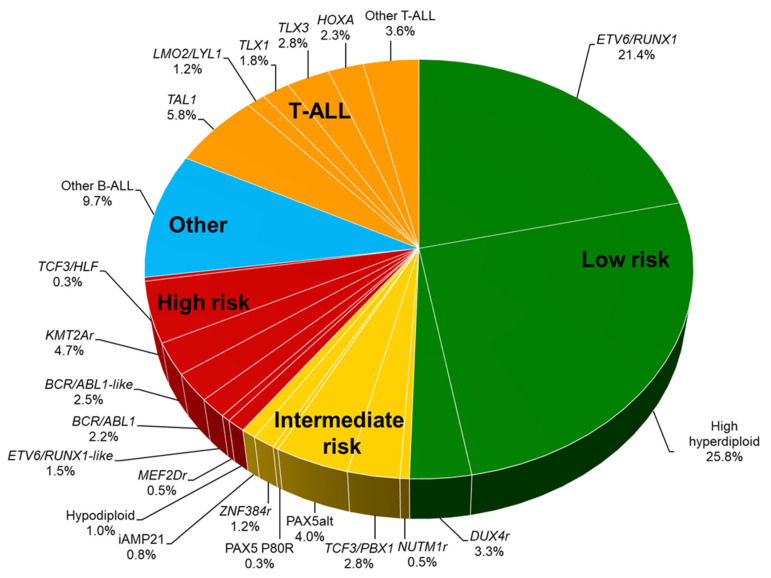
**Distribution of genetic subtypes** Genetic subgroups are listed based on the patients treated in St. Jude Total Therapy Study XVI and on patients with T-ALL who were treated in Children’s Oncology Group studies and evaluated for genetics as part of the Therapeutically Applicable Research to Generate Effective Treatments initiative [11,12]. Percentages are the approximate incidence in pediatric ALL. B-ALL is categorized as low-, intermediate-, or high-risk disease. For T-ALL, no genetic subtypes are clearly associated with outcomes, but the group as a whole is considered an intermediate-risk group. Abbreviations: ALL, acute lymphoblastic leukemia.

**Figure 3 jcm-10-01926-f003:**
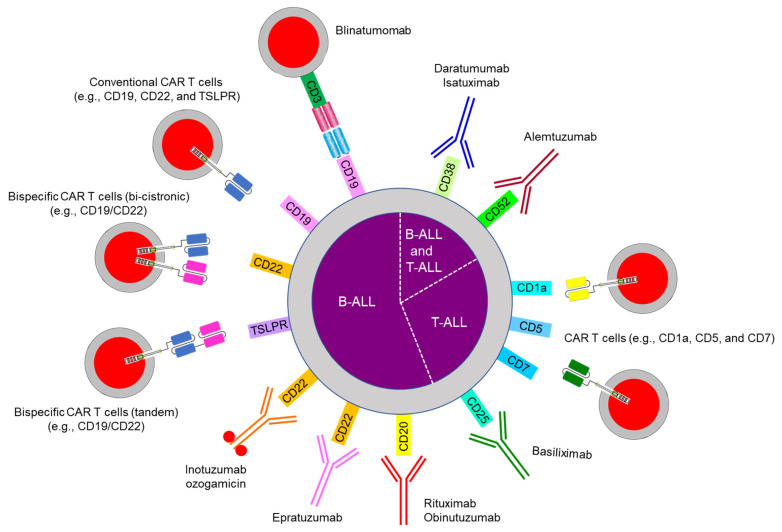
**Immunotherapy in acute lymphoblastic leukemia.** Abbreviations: ALL, acute lymphoblastic leukemia; CAR, chimeric antigen receptor; TSLPR, thymic stromal lymphopoietin receptor.

**Table 1 jcm-10-01926-t001:** Representative medications used in the treatment of patients with acute lymphoblastic leukemia and the year of US Food and Drug Administration approval.

Drugs	Year Approved in the US *
**Conventional chemotherapy**	
Mercaptopurine	1953
Methotrexate	1953
Prednisone	1955
Dexamethasone	1958
Cyclophosphamide	1959
Vincristine	1963
Thioguanine	1966
Cytarabine	1969
Doxorubicin	1974
L-Asparaginase	1978
Daunorubicin	1979
***New formulations or agents***	
Pegaspargase	1994
Nelarabine	2005
Erwinase	2011
Vincristine sulfate liposome injection	2012
Calaspargase	2018
**Molecular targeted therapy**	
***ABL1 inhibitors***	
Imatinib	2001
Dasatinib	2006
Nilotinib	2007
Ponatinib	2012
***JAK inhibitor***	
Ruxolitinib	2011
***BCL-2 and BCL-X_L_ inhibitors***	
Venotoclax	2016
Navitoclax	NA
***Proteasome inhibitors***	
Bortezomib	2003
Carfilzomib	2012
Ixazomib	2015
***mTOR inhibitors***	
Sirolimus	1999
Temsirolimus	2007
Everolimus	2009
***DNA methyltransferase inhibitors***	
Azacitidine	2004
Decitabine	2006
***Histone deacetylase inhibitors***	
Vorinostat	2006
Panobinostat	2015
***Bromodomain inhibitor***	
JQ1	NA
***DOT1 inhibitor***	
Pinometostat	NA
***Menin inhibitor***	
SNDX-5613	NA
**Immunotherapy**	
***Unconjugated antibodies***	
Rituximab (CD20)	1997
Ofatumumab (CD20)	2009
Epratuzumab (CD22)	NA
Daratumumab (CD38)	2015
Alemtuzumab (CD52)	2001
***Bispecific antibody***	
Blinatumomab (CD19)	2014
***Chimeric antigen receptor (CAR) T cells***	
Tisagenlecleucel (CD19)	2017
***Antibody–drug conjugate***	
Inotuzumab ozogamicin (CD22)	2017

* Approval by the US Food and Drug Administration is not limited to indications for pediatric acute lymphoblastic leukemia. Abbreviation: NA, not approved.

**Table 2 jcm-10-01926-t002:** Genetic subtypes and treatment approach.

Category	Characteristics	Therapeutic Approach
**B-lymphoblastic leukemia**		
**Low-risk genetics**		
*ETV6*/*RUNX1*	Excellent prognosis	Reduction of intensity, MRD based
Hyperdiploidy	Excellent prognosis	Reduction of intensity, MRD based
*DUX4*-rearranged	Most have focal *ERG* deletions and favorable outcome despite *IKZF1* alterations	Standard dose intensity, MRD based
**Intermediate-risk genetics**		
*TCF3*/*PBX1*	Higher incidence in African Americans, cytoplasmic μ-chain	Standard dose intensity, MRD based, intensive intrathecal therapy
PAX5alt	*PAX5* fusions, mutation, or amplifications	Standard dose intensity, MRD based
PAX5 p.Pro80Arg	Frequent signaling pathway alterations	Standard dose intensity, MRD based, JAK inhibitors
*ZNF384*-rearranged	Peak age and prognosis vary by fusion partner, expression of myeloid markers	Standard dose intensity, MRD based
iAMP21	Additional copies of chromosome 21, worse outcome with low-intensity therapy	Intensification of therapy
*NUTM1*-rearranged *	Rare; more common in infants, excellent prognosis	Standard dose intensity, MRD based
**High-risk genetics**		
Near-haploid	24–31 chromosomes, Ras-activating mutations, inactivation of *IKZF3*	Intensification of therapy, MRD based, BCL-2 inhibitors
Low-hypodiploid	32–39 chromosomes, *TP53* mutations (somatic and germline)	Intensification of therapy, MRD based, BCL-2 inhibitors
*BCR*/*ABL1*	Prognosis improved with ABL1 inhibitors, common deletions of *IKZF1*	ABL1 inhibitors, BCL-2 inhibitors
*BCR*/*ABL1*-like; JAK-STAT activating mutation	*CRLF2* rearranged (*IGH*-*CRLF2*, *P2RY8*-*CRLF2*), *JAK1/2, EPOR, IL7R, SH2B3* mutation	JAK inhibitors, BCL-2 inhibitors
*BCR*/*ABL1*-like; *ABL1*-class	Kinase-activating lesions, potentially amenable to kinase inhibition	ABL1 inhibitors, BCL-2 inhibitors
*KMT2A* (*MLL*)-rearranged	Common in infant ALL, few cooperating mutations	DOT1L inhibitors, menin inhibitors, proteasome inhibitors, histone deacetylase inhibitors, BCL-2 inhibitors
*MEF2D*-rearranged	Mature B cell leukemia morphology, cytoplasmic μ-chain	Histone deacetylase inhibitors, proteasome inhibitors
*TCF3-HLF*	Rare; dismal prognosis	BCL-2 inhibitors
*ETV6/RUNX1*-like *	Similar gene expression profile to *ETV6*-*RUNX1* but lacks fusion	Intensification of therapy, MRD based
**T-lymphoblastic leukemia**		
Non-early T-cell precursor	Deregulation of *TAL1*, *TAL2*, *LYL1*, *LMO1*, *LMO2*, *TLX1* (*HOX11*), *TLX3* (*HOX11L2*), and *HOXA*; *NOTCH1* activating mutation	Standard dose intensity, MRD based, nelarabine, BCL-2 inhibitors
JAK-STAT activating mutation	Approximately 25% of patients with T-ALL	Standard dose intensity, MRD based, nelarabine, JAK inhibitors, BCL-2 inhibitors
*ABL1* fusions (e.g., *NUP214*-*ABL1*)	Fusion with *BCR* and *NUP214,* potentially amenable to tyrosine kinase inhibition	Standard dose intensity, MRD based, ABL1 inhibitors, nelarabine, BCL-2 inhibitors
Early T-cell precursor ALL	Mutations in transcriptional regulators, JAK-STAT and Ras signaling, and epigenetic modifiers	Standard dose intensity, MRD based, JAK inhibitors, BCL-2 inhibitors

* Newly identified subgroups, necessary to confirm their prognosis in a larger number of patients. Abbreviations: MRD, minimal residual disease; iAMP21, intrachromosomal amplification of chromosome 21; ALL, acute lymphoblastic leukemia.

## Data Availability

Data sharing not applicable.

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
