# Peer review of "Advances in the Diagnosis and Treatment of Pediatric Acute Lymphoblastic Leukemia"

_jcm, 2021, doi:10.3390/jcm10091926_

Round 1

Reviewer 1 Report

The manuscript is very well written and comprehensive of most advanced biologic features and treatment options in childhood ALL.

I have minimal suggestions.

1) Legend Fig 2: Total Therapy. Study XVI: dot is a typo error.

2) Lines 87-91: it should be specified that dose reduction was randomized only in the delayed intensification phase in the study mentioned.

3) Line 119: MRD was measured at day 19 (not 15) in these patients, correct?

4) Line 166: it should be mentioned also that the COG AALL0622 study with dasatinib given at 60 mg/sqm did not show any improvement when compared with the previous COG AALL0031 study where imatinib was given on top of the same chemotherapy. (Slayton WB, JCO 2018)

Author Response

The manuscript is very well written and comprehensive of most advanced biologic features and treatment options in childhood ALL.

I have minimal suggestions.

1) Legend Fig 2: Total Therapy. Study XVI: dot is a typo error.

Response: Thank you. The dot has been removed.

2) Lines 87-91: it should be specified that dose reduction was randomized only in the delayed intensification phase in the study mentioned.

Response: Thank you. We have clarified this.

3) Line 119: MRD was measured at day 19 (not 15) in these patients, correct?

Response: Thank you. This has been fixed.

4) Line 166: it should be mentioned also that the COG AALL0622 study with dasatinib given at 60 mg/sqm did not show any improvement when compared with the previous COG AALL0031 study where imatinib was given on top of the same chemotherapy. (Slayton WB, JCO 2018)

Response: This paper by Slayton et al. is now cited as reference 43, and the finding has been described.

Reviewer 2 Report

This is a comprehensive, updated, informative and pleasant review about pediatric ALL. 

I have only a minimal suggestion: when speaking about efficacy of TKI in ABL-class rearranged BCR-ABL1-like ALL (lines 214 and 434) consider citing the work by Tanasi et al. Blood 2019, which provides information about the efficacy of BCR-ABL inhibitors in this group of patients, including some pediatric cases.

Author Response

This is a comprehensive, updated, informative and pleasant review about pediatric ALL. 

I have only a minimal suggestion: when speaking about efficacy of TKI in ABL-class rearranged BCR-ABL1-like ALL (lines 214 and 434) consider citing the work by Tanasi et al. Blood 2019, which provides information about the efficacy of BCR-ABL inhibitors in this group of patients, including some pediatric cases.

 Response: Thank you. This paper has been cited as reference 55.

Reviewer 3 Report

Advances in the Diagnosis and Treatment of Pediatric Acute Lymphoblastic Leukemia

The manuscript is well written and highlights the recent advances in acute lymphoblastic leukemia (ALL), focusing also on some recent molecular targeted therapies.

The manuscript is in a correct and understandable English and therefore does not require revision for English.

I would have some minor revisions:

I would suggest expanding a little more the introduction, adding a more detailed description of ALL and difference between T- and B-ALL, distinction that will be more fully described in the following next text paragraphs.

I would also expand the paragraph on targeted therapies. Table 1 lists some relevant drugs as small molecule inhibitors, such as mTOR inhibitors. Given the importance of the PI3K / Akt / mTOR pathway in leukemias, I would add a paragraph focused on this pathway by inserting some scientific evidence of the most recent inhibitors, including Sirolimus, Temsirolimus and Everolimus. Similarly, I would describe demethylating agents, as well as histone deacetylase inhibitors.

There are no other comments to add.

Best regards

Carolina Simioni

Author Response

The manuscript is well written and highlights the recent advances in acute lymphoblastic leukemia (ALL), focusing also on some recent molecular targeted therapies.

The manuscript is in a correct and understandable English and therefore does not require revision for English.

I would have some minor revisions:

I would suggest expanding a little more the introduction, adding a more detailed description of ALL and difference between T- and B-ALL, distinction that will be more fully described in the following next text paragraphs.

Response: Thank you. We have expanded the introduction by explaining the epidemiology of ALL.

I would also expand the paragraph on targeted therapies. Table 1 lists some relevant drugs as small molecule inhibitors, such as mTOR inhibitors. Given the importance of the PI3K / Akt / mTOR pathway in leukemias, I would add a paragraph focused on this pathway by inserting some scientific evidence of the most recent inhibitors, including Sirolimus, Temsirolimus and Everolimus. Similarly, I would describe demethylating agents, as well as histone deacetylase inhibitors.

Response: Although the clinical experience in pediatric ALL is limited, we have added some discussion of recent studies on mTOR inhibitors, demethylating agents, and histone deacetylase inhibitors.

There are no other comments to add.

Best regards

Carolina Simioni

Reviewer 4 Report

1. For section 4.2 (ph + ALL), it may be worth adding a sentence referring to the data from the EWALL study (Rousselot et al, Blood 2016), where Dasatinib with low intensity chemotherapy was used in elderly patients. this study has longer follow up data, with 5yr OS of 36-45%

2. for section 7.1 (ETP ALL), consider adding data from Jain et al, Blood 2016, regarding lower outcomes in ETP compared to non-ETP in adults treated at MDACC with 3 different regimens. 

3. for section 10.1 (Blina), consider adding data to show relapses are related primarily to antigen loss. 

4. For section 10.2 (CAR-T) line 508, regarding infection risk, please refer to "how I prevent infections in CD19 targeted CAR-T...." by Hill et al in Blood 2020, with recommendations for prophylaxis. PJP prophylaxis should continue for at least 6 months. Data from Baird et al, Blood advances 2020 suggest 18 months Zoster and PJP prophylaxis. 

5. Line 520, please add reference to paper by Pan et al, Blood 2020 regarding sequential therapy with CD19 and CD22 CAR. 

6. section 10.3, regarding recommendations for prevention of VOD, please refer to paper by Kebriaei et al, Bone Marrow Transplant, 2018. limit Ino to 2 cycles if HCT is planned, avoid dual alkylating agents for conditioning, use prophylactic Urso. some groups, including ours, advise a 2 month washout period after Ino and before transplant conditioning. 

Author Response

  1. For section 4.2 (ph + ALL), it may be worth adding a sentence referring to the data from the EWALL study (Rousselot et al, Blood 2016), where Dasatinib with low intensity chemotherapy was used in elderly patients. this study has longer follow up data, with 5yr OS of 36-45%

Response: Thank you very much for this suggestion. As we are primarily describing pediatric ALL, we prefer to focus on pediatric studies. Therefore, instead of the EWALL study, we have cited the COG AALL0622 study as suggested by reviewer 1.

  1. for section 7.1 (ETP ALL), consider adding data from Jain et al, Blood 2016, regarding lower outcomes in ETP compared to non-ETP in adults treated at MDACC with 3 different regimens. 

Response: We have mentioned the worse outcomes in ETP ALL, as compared with non-ETP ALL, and we have cited the paper by Jain et al. as reference 94.  

  1. for section 10.1 (Blina), consider adding data to show relapses are related primarily to antigen loss. 

Response: This mechanism is now described.

  1. For section 10.2 (CAR-T) line 508, regarding infection risk, please refer to "how I prevent infections in CD19 targeted CAR-T...." by Hill et al in Blood 2020, with recommendations for prophylaxis. PJP prophylaxis should continue for at least 6 months. Data from Baird et al, Blood advances 2020 suggest 18 months Zoster and PJP prophylaxis. 

Response: Thank you very much. The paper by Hill et al. is cited as reference 140.

  1. Line 520, please add reference to paper by Pan et al, Blood 2020 regarding sequential therapy with CD19 and CD22 CAR. 

Response: Thank you. This paper is now cited as reference 147.

  1. section 10.3, regarding recommendations for prevention of VOD, please refer to paper by Kebriaei et al, Bone Marrow Transplant, 2018. limit Ino to 2 cycles if HCT is planned, avoid dual alkylating agents for conditioning, use prophylactic Urso. some groups, including ours, advise a 2 month washout period after Ino and before transplant conditioning. 

Response: Thank you very much. The paper by Kebriaei et al. is cited as reference 151, and the issues mentioned have been described.